# Induction of Male Sterility by Targeted Mutation of a Restorer-of-Fertility Gene with CRISPR/Cas9-Mediated Genome Editing in *Brassica napus* L.

**DOI:** 10.3390/plants11243501

**Published:** 2022-12-13

**Authors:** Zunaira Farooq, Muhammad Nouman Riaz, Muhammad Shoaib Farooq, Yifan Li, Huadong Wang, Mayra Ahmad, Jinxing Tu, Chaozhi Ma, Cheng Dai, Jing Wen, Jinxiong Shen, Tingdong Fu, Shouping Yang, Benqi Wang, Bin Yi

**Affiliations:** 1National Key Laboratory of Crop Genetic Improvement, National Center of Rapeseed Improvement, Huazhong Agricultural University, Wuhan 430070, China; 2Soybean Research Institute, National Center for Soybean Improvement, Key Laboratory of Biology and Genetic Improvement of Soybean (General Ministry of Agriculture), Jiangsu Collaborative Innovation Center for Modern Crop Production, Nanjing Agricultural University, Nanjing 210095, China; 3Wuhan Vegetable Research Institute, Wuhan Academy of Agricultural Science and Technology, Wuhan 430065, China

**Keywords:** rapeseed CMS, *Rf*-like (*RFL*), CRISPR/Cas9, sgRNA, cytological study, genome editing

## Abstract

*Brassica napus* L. (canola, oil seed rape) is one of the world’s most important oil seed crops. In the last four decades, the discovery of cytoplasmic male-sterility (CMS) systems and the restoration of fertility (*Rf*) genes in *B. napus* has improved the crop traits by heterosis. The homologs of *Rf* genes, known as the restoration of fertility-like (*RFL*) genes, have also gained importance because of their similarities with *Rf* genes. Such as a high non-synonymous/synonymous codon replacement ratio (dN/dS), autonomous gene duplications, and a possible engrossment in fertility restoration. *B. napus* contains 53 *RFL* genes on chromosomes A9 and C8. Our research aims to study the function of *BnaRFL11* in fertility restoration using the CRISPR/Cas9 genome editing technique. A total of 88/108 (81.48%) T_0_ lines, and for T_1_, 110/145 (75%) lines carried T-DNA insertions. Stable mutations were detected in the T_0_ and T_1_ generations, with an average allelic mutation transmission rate of 81%. We used CRISPR-P software to detect off-target 50 plants sequenced from the T_0_ generation that showed no off-target mutation, signifying that if the designed sgRNA is specific for the target, the off-target effects are negligible. We also concluded that the mutagenic competence of the designed sgRNAs mediated by U6-26 and U6-29 ranged widely from 31% to 96%. The phenotypic analysis of *bnarfl11* revealed defects in the floral structure, leaf size, branch number, and seed production. We discovered a significant difference between the sterile line and fertile line flower development after using a stereomicroscope and scanning electron microscope. The pollen visibility test showed that the pollen grain had utterly degenerated. The cytological observations of homozygous mutant plants showed an anther abortion stage similar to nap-CMS, with a *Orf222*, *Orf139*, *Ap3*, and *nad5c* gene upregulation. The *bnarfl11* shows vegetative defects, including fewer branches and a reduced leaf size, suggesting that PPR-encoding genes are essential for the plants’ vegetative and reproductive growth. Our results demonstrated that *BnaRFL11* has a possible role in fertility restoration. The current study’s findings suggest that CRISPR/Cas9 mutations may divulge the functions of genes in polyploid species and provide agronomically desirable traits through a targeted mutation.

## 1. Introduction

Rapeseed is a globally important crop that provides ingestion products for humans and animal feed, such as oil products, proteins, and raw materials for industrialized products worldwide [1,2]. Male sterility is one of the most effective heterosis techniques for increasing crop production. Maternal inheritance is essential for hybrid breeding programs and mitochondrial and chloroplast genomes. CMS is a maternally inherited trait in flowering plants that cannot produce functional pollen [3]. Several cellular metabolic processes mandatory for higher plants, such as ATP production by oxidative phosphorylation, occur in mitochondria. Due to a mitochondrial impairment, cytoplasmic male sterility offers unique mechanisms for revealing the genetic association between nuclear genomes and mitochondria in plants [4]. The most common male sterility induction mechanisms in rapeseed are GMS, CMS, and chemical gametocide [5]. In many plants, CMS traits are determined by *ORFs* encoded by the mitochondrial genome. Typically, rapeseed has two CMS mitotypes, nap and Pol. According to previous studies, the CMS in rapeseed is of the following types: *nap* CMS [6], *ogu* CMS [7], *pol* CMS, Moricandia arvensis [8], *shan2A* [9], Tour CMS [10], *Nca* CMS [11], *oxa* CMS [12], *Nsa* [13], *inap* CMS, and *hau* CMS [14].

The effects of mitochondrial genes controlling cytoplasmic male sterility (CMS) can be suppressed by fertility restoration genes (*Rf*). Several *Rf* genes in rapeseed and other crops encode the pentatricopeptide repeat protein (PPR) family [15]. PPR proteins are a group of pattern-specific single-stranded RNA-associated proteins found in terrestrial plants that regulate post-transcriptional mechanisms, such as RNA degradation, cleavage, stability, splicing, and genome editing in mitochondrial and chloroplast genomes [16].

According to previous research, the *Rf* genes found in petunia [17], rice [18,19], radish [20,21], and sorghum [22] are members of the pentatricopeptide repeat gene family. In *Brassicaceae,* crops have multiple fertility restorer genes (*Rf)* encoding PPR proteins such as Nap-*orf222-Rfn* [23], Pol-*orf224-Rfp* [23], Ogu-*orf138-Rfo* [24], Tour-*orf263-Rf1_Rf2_* [25,26], Kos-*orf125-Rfk* [24,27], Nsa-*orf224,orf309*, *orf346-*unknown [28], and Shaan2a-*orf224-1-Shaan2b* [29]. Previous studies have revealed that both *Rf*-CMS genes interact throughout evolution, which may explain why all cloned *Rf*-PPR genes share a common ancestor [30,31].

According to several studies, *Rf-PPR* genes are also recruited to mitochondria, which block CMS gene products’ accretion [32]. Additionally, many *Rf* genes encoding PPR proteins are invariably present in clusters of linked *Rf-*PPR*-*like genes (*RFL*) [33,34]. *RFL*-rich areas contain many *Rf-*PPR genes, and it is hypothesized that most *RFL* genes in the same chromosomal region are functional restorer genes. For instance, the rice chromosome 10 *RFL*-rich area contains *Rf1* and *Rf4* [35,36]. The *Rf8* locus in maize is located in an *RFL* cluster on chromosome 2 [37].

Fortunately, recent research has revealed that 53 *RFL* genes have been discovered in *Brassica napus*. It has been demonstrated that 10 and 18 *BnaRFLs* assemble into highly dense clusters on chromosomes A9 and C8, respectively. These nominated restorer *BnaRFLs* on chromosomes A9 and C8 likely play a conserved role in mitochondrial RNA processing. It can be concluded that six *BnaRFLs* are candidates for fertility restorer genes (*RFL3*, *RFL4*, *RFL5*, *RFL8*, *RFL15*, and *RFL41*), four genes are similar to restorer genes (*RFL2*, *RFL10*, *RFL11,* and *RFL42*), and two restore genes (*Rfn-RFL6* and *Rfp-RFL13*) clustered together in the phylogenetic tree, indicating that these genes were the most likely restorer gene members in the CMS rapeseed model. When comparing *BnaRFL* to other known restorer genes, it was discovered that all the identified *BnaRFLs* had 80 amino acid motifs*, suggesting* that *BnaRFL* could help in a rapeseed fertility restoration. It indicates that the *RFL* genes revealed in rapeseed have a possible role in fertility restoration on the A9 and C8 chromosomes [29].

Furthermore, they discovered *BnRFL13* (*Rfp*) and *BnRFL6* (*Rfn*), which have already been proven to be restorer genes in Pol and nap CMS in rapeseed [37,38,39]. *RFL* genes, grouped on chromosomes A9 and C8, have been identified as restorer genes in the rapeseed CMS system. These genes have a significant impact on mitochondrial RNA processing [29].

Consequently, we can learn how putative *RFL* genes restore rapeseed fertility. The *CRISPR/Cas9* genome editing method has recently been extensively used in rapeseed to develop *BnTT8* plants with a double mutant phenotype, displaying increased oil and protein levels with a changed fatty acid (F.A.) content but no major yield defects [40]. In the current study, we used this technique to develop a *BnaRFL11* mutant to induce mutations by knocking out the *RFL* gene, a candidate for fertility restoration in rapeseed CMS.

## 2. Material and Methods

### 2.1. Plant Materials

In the present study, a pure line of *B. napus* Westar (nap-cytoplasm) was used as the source of transformation. The seeds were obtained from the National Engineering Research Center of Rapeseed (Wuhan, China). Wild-type plants and transgenic lines were grown in a greenhouse under (16/8 h (h) light/dark cycle at 22 °C).

### 2.2. Phylogenetic and Bioinformatics Analysis

All *Brassica napus* selected gene sequences, and other species’ homologs, were subjected to Mega 7 (https://www.megasoftware.net/ accessed on 20 May 2019) for phylogenetic analysis using the neighbor-joining method. The sequences for the trees that only contained *Brassica* and *Arabidopsis* were downloaded from BRAD and TAIR, respectively, and conserved motifs were identified using the MEME online suit (http://meme-suite.org/tools/meme, accessed on 25 May 2019) with the default settings [41].

### 2.3. Computational sgRNA Design and Selection

The sgRNAs were designed using the CRISPR-P web tool (http://cbi.hzau.edu.cn/cgi-bin/CRISPR accessed on 25 May 2019) [42]. First, we found all possible sgRNA sequences in specific target genes with a G.C. content (45~60%) and an on-target score (score value > 0.6). The off-target scores were then calculated based on a previous study’s scoring system of off-target sites [43].

### 2.4. Construction of Binary Vectors and Genetic Transformation of Plants

To construct Cas9/sgRNA-expressing binary vectors, variant sgRNAs with a sequence specificity were designed for the gene of interest using the web-based tool CRISPR-P (http://cbi.hzau.edu.cn/cgi-bin/CRISPR, accessed on 25 May 2019). For constructing vectors using a multiplex genome targeting system, the binary pHSE401-2gR/Cas9 was arranged by Prof. Wang Zhiping (Chen Qijun, College of Biology, China Agricultural University China) and used to construct a library according to the findings of [44]. The desired constructs were monitored by sequencing and finally transformed into *B. napus* using the Agrobacterium-mediated hypocotyl method, as reported previously [45]. The transgenic lines were confirmed using a polymerase chain reaction (PCR) and antibiotic selection markers. The oligo primers used for preparing the sgRNA vectors are listed in (Appendix A).

### 2.5. Identification of Transgenic Mutant Plants and Potential Off-Targets

The presence of the T-DNA construct was assessed by PCR using CPA F/R gene-specific primers. A PCR was performed to amplify the genomic region surrounding the CRISPR target sites using specific primers (Appendix A), and the mutations were screened using the PAGE method previously described [46]. To confirm the PAGE-based genotyping results, we used the high-throughput tracking of mutations via the T.A. cloning of the PCR products. The pMD18-T vector (Takara) was used for T.A. cloning and sequencing. The potential off-target sites were identified using CRISPR-P (http://cbi.hzau.edu.cn/cgi-bin/CRISPR, accessed on 5 June 2020). The PCR amplified an approximately 250-bp DNA sequence covering each off-target site. The primers used are listed in (Appendix A). For each target gene, the mixed genomic DNA from T_0_-edited plants was used as the template, and wild-type DNA was included as a control. All the PCR products were purified and mixed in equal amounts (50 ng each) as a single sample. The DNA library construction and sequencing were performed using the T.A. cloning method.

### 2.6. RNA Extraction, RT-PCR

The total RNA was extracted by Trizol reagent (Invitrogen, Waltham, MA, USA). The total RNA (5 μg) was treated with DNase (Thermo Fermentas, Waltham, MA, USA), purified, and precipitated using ethanol. The cDNA was obtained by reverse transcription using Superscript III according to the manufacturer’s instructions. The SYBR Green I master PCR kit was used for real-time PCR with gene-specific primers on a Light Cycler 480.

### 2.7. Histological Analyses

#### 2.7.1. Scanning Electron Microscopy Analysis

Fresh WT (Westar) and mutant plant pollen grains were air-dried for approximately 30 min before viewing with a Hitachi S 4700 scanning electron microscope at an accelerating voltage of 5 kV.

#### 2.7.2. Pollen Fertility Examination

The pollen sampled from the W.T. and mutant plants’ rapeseed buds immediately before flowering were stained with 1% (*w*/*v*) acetocarmine solution to analyze the pollen viability. Three biological replicates were used for this study. The stained pollen grains were visualized, and the images were recorded using a Leica DMIRB fluorescence microscope.

#### 2.7.3. Cytological Analysis

The sections of flower buds from the CMS and fertile plants were obtained following the method described in [47]. The sample was embedded and sectioned at a 6 to 10 μm thickness using a Leica 2035 Biocut. The sections were stained with hematoxylin and eosin (0.5%) for the anthers. The images were captured using a Nikon Eclipse 80i microscope.

### 2.8. Field Experiments and Phenotyping

Wild-type plants and T_0_ and T_1_ transgenic lines were kept in a greenhouse under a photoperiod (16/8 h light/dark at 22 °C) in 2019 and 2020. After screening, the homozygous mutant lines were grown during the rapeseed crop season (2020–2021) in the experimental field of Huazhong Agriculture University, Wuhan, China. The experiment was performed on field plants, followed by three replicates with a complete block design. In one row, approximately 11–12 plants were planted; each row followed the same procedure, with a distance observed in one row of 21 cm, followed by a 30 cm distance in each row. The field management of each line was performed using a standard breeding method.

## 3. Results

### 3.1. Bioinformatic Analysis of BnaRFL11 Gene in Westar

According to the genome information, *B. napus* contains one copy of *BnaRFL11*. (http://www.genoscope.cns.fr/brassicanapus/, accessed on 20 May 2019). The DNA and protein sequence alignments were similar for a single copy of *the BnaA09g45590D,* At*1g12300,* and *At1g12620* genes (Appendix A). *BnaRFL11 and Arabidopsis* homologs DNA sequence were given in to MEME “(http://meme-suite.org/, accessed on 25 May 2019)” to search for the conserved motifs between the selected gene protein sequences (Figure 1b). According to the output, all the motifs were highly conserved in these DNA sequences, suggesting a similarity in function (Figure 1b). The phylogenetic tree showed that a single copy of *BnaRFL11* was highly similar to its homologs in *A. thaliana* and *Rfn* (*RFL6*) (Figure 1a).

### 3.2. CRISPR/Cas9 Vector Construction to Knock out the BnaRFL11 Gene in Rapeseed

Many fertility restorer genes have been reported to encode the PPR protein family essential for a fertility restoration in petunia, radish, rice, *Arabidopsis*, and *Brassica* species [48,49]. Thus, the modification of *PPR-encoding* genes may induce male sterility in rapeseed. We used the pure Westar line of *B. napus,* which is amenable to an Agrobacterium-mediated transformation. A single copy of *BnaRFL11* (BnaA09.PPR.AA09) has six exons, similar to the *Arabidopsis* sister genes. Two sgRNA (S1–S2) were designated, employing the “CRISPR-P” [44] to generate point mutations in a single copy of *BnaRFL11* using a Cas9 gene-editing technique (Figure 2 and Appendix A). S1 and S2 are located on the different exons of the *BnaRFL11* gene.

We used a multiplex genome-editing technique with a single CRISPR/Cas9 vector. The CRISPR vector based on pCAMBIA was arranged with these two sgRNAs with Cas9 driven by U6-26p as the promoter of the *Arabidopsis* U6 gene; U6-26t is the terminator of the U6-26 gene with the downstream sequence. The CaMV35S promoter was used for the Cas9 protein expression analysis. In addition, the U6-26p and U6-29p promoters were used to drive S1 and S2 from *Arabidopsis* [44] (Figure 2b). Such a design will determine which promoter is best for generating an adequate mutation in the desired gene sequence with the Cas9 protein expression in *B. napus.*

### 3.3. Identification of Mutation Patterns in the Transgenic Plants

The *BnaRFL11* CRISPR/Cas9 vector was positively transferred into Westar by an Agrobacterium transformation, and 108 independent lines were developed for *bnarfl11* (Figure 3). To check the positive transgenic plants through a PCR using the construct-specific primers, 88/108 (81.48%) of the T_0_ lines carried T-DNA insertions, and for T_1_, 110/145 (75%) lines carried T-DNA insertions (Table 1). Polyacrylamide gel electrophoresis (PAGE) was used to identify the edited lines. Following the standard procedure, the PCR products from each target site were denatured, renatured, and separated. This method is based on the slower migration of heteroduplex DNA (mutation) than of homoduplex DNA (non-mutation) on native PAGE [46]. The genotyping results of many plants displayed band profiles that differed from those on the non-denaturing PAGE gels in wild-type plants (W.T.), showing the presence of mutations at the target sites (Appendix A).

We used a high-throughput tracking system of mutations, the T.A. cloning of PCR products (gel recovery and ligation of DNA), to detect T_0_ and T_1_ positive transgenic plant mutations to authenticate the PAGE results. The site-specific primers for each target were designed based on the SNP differences. For each target, forward and reverse primers covered the target site, and both primers were designed within 20–80 nucleotides from the target site (Appendix A).

Transgenic plants were sequenced for each targeted area following T.A. cloning to calculate the mutation rate correctly. The sequencing chromatograms of each line were manually examined. When the sequencing chromatograms demonstrated a nucleotide substitution (deletion, insertion, or substitution) or several indications (overlapping peaks) at the sgRNA target locations, we concluded that mutations were successfully induced (Appendix A), indicating that T. A. clone-based sequencing is an effective and valuable technique for identifying plant mutations. CRISPR/Cas9 induces mutations in the designed S1 and S2 targets in a single copy of the *BnaRFL11*. Of the 88 transgenic plants, a total of 70 T_0_ mutant lines were examined; 25/70 (35.71%) loci of *bnarfl11* showed putative heterozygous (Hetro) alterations, 7/70 (10%) loci had chimeric changes, and more than half 38/70 (54.28%) of the loci were homozygous (Homo) mutations. Mutant lines with single and quadruple mutations were selected for the genotyping and phenotypic analysis. Seventeen mutant lines were selected from the 70 for the T_1_ generation for a further confirmation (Appendix A).

### 3.4. Isolation of Mutants with Transgenic Elements in T_0_ and T_1_ Generations

To develop stable mutant lines, 17 T_0_ lines carrying mutations and a male-sterile phenotype in *bnarfl11* were crossed with the wild type to obtain the seeds, and an individual T_1_ generation was genotyped through the T.A. clone. The observed transmission rate of the allelic mutations from T_0_ mutant plants to T_1_ progenies ranged from 35% to 100% in *bnarfl11,* but the average transmission rate from T_0_ to T_1_ plants with allelic mutations was 81% (Appendix A). For example, the detected mutations in the T_1_ generation matched the observed mutations in the related T_0_ lines (Appendix A). Homozygous mutations were detected in the T_1_ plants generated at more than two loci, indicating that the next-generation mutant was stably inherited and permanently exhibited homozygous genotypes (Appendix A). Interestingly, one and two new mutations were identified in the T_1_ lines.

In contrast, these mutations were not present in T_0_ lines at the S2 site of *bnarfl11* (Appendix A), indicating that wild-type sites were modified during the development of independent lines with a low efficiency. Consistently, the observed mutation and male sterility phenotypes in the corresponding *bnarfl11-1*, *bnarfl11-5*, and *bnarfl11-8* T_0_ plants matched the detected changes and detected phenotypes in the next T_1_ (*bnarfl11-1-5*, *bnarfl11-5-8*, and *bnarfl11-8-3*) progeny lines (Appendix A and Figure 4). Thus, the sequencing data of the heritable changes at both target sites of *bnarfl11* (*BnaA09g45590D*) after a T_0_ to T_1_ generation provides strong evidence that the constant Cas9-induced germline mutation transfer of mutations in *B. napus* is achievable (Figure 4). To screen the mutants with a targeted modification without integrating foreign DNA into the *B. napus* genome, we performed a PCR analysis of the T_0_ and T_1_ generations using Cas9-888 and U6-26p primers to confirm the positive transgenic plants. The Cas9 vector with T-DNA was not detected in 20 of 108 (18.51%) T_0_ plants and 35/145 (24.1%) T_1_ plants originating from 17 independent T_0_ lines (Table 1). Altogether, various *bnarfl11* single, double, triple, and quadruple homozygous transgene-free T-DNA mutants were obtained in the T_1_ generation. Thus, transgene-free T-DNA plants carry out the desired gene modifications that might be obtained through heritable segregation in *B. napus.*

### 3.5. CRISPR’s Off-Target Effects on Mutant Lines (T_0_) in B. napus

In the current study, we used CRISPR-P software to detect off-targets. We searched for putative off-targets with similar identities to the two sgRNAs used for an on-target mutation in *Brassica napus* with the help of “CRISPR-P” [50]. Some off-target genes were identified for the selected sgRNAs (Appendix A). There were four off-targets for S1 with a maximum of four mismatches, whereas S2 had three off-targets with a maximum of four mismatches in *BnaRFL11*. We sequenced 50 plants from the T_0_ generation that showed no off-target mutations, signifying that if the designed sgRNA is specific for the target, the off-target effects are negligible. It also emphasized that the male-sterile phenotype detected in *bnarfl11* was due to an induced mutation in *BnaA09g45590D* and not by off-target effects. Therefore, the CRISPR/Cas9 system can induce stable and specific mutations in the *B. napus* genome.

### 3.6. Editing Efficiency of sgRNAs in BnaRFL11

Different sgRNAs have different mutation rates. Therefore, the two sgRNAs for the gene ensured a high rate of mutation (Appendix A). Interestingly, these sgRNAs created a targeted mutation in a single copy of *BnaRFL11* and developed single, double, triple, and quadrable mutations in the T_0_ generation (Appendix A, Figure 4). The highest sgRNA editing efficiency was recorded at S1 (64.81%) in *bnarfl11* and S2 (46.29%) in *bnarfl11* (Appendix A). This suggests that the practical selection of sgRNAs is crucial for effectively generating mutations in the target sequence. The efficiency of sgRNA depends on the promoters expressing the Cas9 protein in *B. napus.* The mutagenic competence of the designed sgRNAs mediated by U6-26 and U6-29 ranged from 31% to 96% (Appendix A), indicating that not every promoter can effectively mediate genome editing in *B. napus*. This is the first time a highly conserved *bnarfl11* gene has been knocked out in *B. napus* to generate quadrable mutations. The generation of quadrable mutations in a single copy of *bnarfl11* was due to the highly efficient design of sgRNAs.

### 3.7. Morphological Analysis of bnarfl11 Mutant Plants and Pollen Viability Test

For *bnarfl*11 gene-phenotype analysis, all homozygous mutant T_1_ lines were grown in the experimental field of Huazhong Agricultural University, Wuhan, China.

There were no morphological differences between sterile and fertile plants during the shoot and root growth cycles. The morphology of fertile flower buds and flowers of *bnarfl11* differed significantly during the later stages of development. We discovered a significant difference between the sterile line and fertile line anther development after using a stereomicroscope and scanning electron microscopy. The petals shrank and contracted; there was no pollen and much less nectary in the sterile plants, whereas the fertile plant was stable, with wild-type stamen and petal growth and regular nectary. The fertile flowers were larger than the *bnarfl11* flowers. During the growth process, the filaments and anthers of sterile flowers are often smaller than those of fertile flowers. (Figure 5a,b) shows the wild-type side view and upper view with normal anther development (Figure 5c,d). The *bnarfl11* side view and upper view mutant anther development (Figure 5a–d).

Furthermore, sterile anthers produced very diminutive but degenerated pollen and did not affect the pistil growth. For the pollen viability test, staining with a 1% (*w*/*v*) acetocarmine solution showed that *bnarfl11* had no pollen if some flowers succeeded in producing anthers. The pollen visibility test showed that the pollen grains were degenerated entirely (Figure 6a,b). As mentioned in many research articles, the PPR mutant showed many morphological defects; therefore, *bnarfl11* showed vegetative defects, including fewer branches and a reduced leaf size. (Figure 7a–f) shows various stages from bud development to seed production, and sterile plants produced significantly less seed than W.T. while crossing with the wild type to produce pure F1 hybrid seeds with a stable Cas9 vector transmission.

### 3.8. Cytological Analysis of Anther Abortion in bnarfl11

Theis and Robbelen (1990) described all the fertile anther stages of development; according to them, the results were as follows.

#### Development of *bnarfl11* Anthers

There was no structural distinction between fertile and non-fertile individuals in the 1st stage named primordial. Distinctions emerge when primordial tissue is divided into sporogenous, vascular, and parenchymal tissues. During the early stages, the sterile buds lose one to three locks per anther than fertile buds, producing four horizontal, angular locules. In sterile flowers, the number of locules varies from one to another.

Furthermore, in sterile buds, the development of locules inside an anther has polarity: adaxial locks often grow, while other locks are mostly sterile. The tapetum is frequently dense and disconnected from the microspore mother cells (MMC) of growing CMS locules. Surprisingly, this has little effect on their inconsistency: sexual cell division occurs in locules, and Tds segregate into immature pollen in the same way as fertile lines do. Both locules inside and between the fertile anthers mature simultaneously, going through various stages of development simultaneously. The fusion of two adaxial locks and the early displacement of a tapetum from its locule layer are typical defects in CMS anthers. Rather than dividing fibrous tissue throughout the MMC stage, the tapetum in sterile anthers is frequently distinguished during the MMC process. The pollen grains that grew within the formed locules grew as fertile pollen but became degenerated, considering the numerous structural variations detected in the CMS anthers (Figure 8a–p and Figure 9a–d).

### 3.9. The Behavior of nap-CMS Causing Gene in bnarfl11

First, we checked the cytoplasm type using the specific primers designed by [23] (Appendix A) and found that the cytoplasm type was nap-cytoplasm (Figure 10a). Therefore, we can say that the sterility type observed was nap-CMS. To determine the expression of the nap-CMS-causing gene, we performed qRT-PCR. Because *AP3* is essential for determining the individuality and behavioral symmetry of petals and stamens, its expression was significantly high in buds and petals. Furthermore, the expression of *orf222*, *orf139*, and *nad5c* was significantly upregulated in the stamens and petals in the sterile bud compared to other organs of plants and the wild type (Figure 10b–f).

## 4. Discussion

### 4.1. In B. napus, the CRISPR/Cas9 System Is Highly Effective in Producing Targeted Mutations

Since it is easy to produce mutations in single genes or various genetic sequences with unknown functions, the CRISPR/Cas9 system has enormous potential for facilitating plant functional studies [51]. However, the challenge of rapidly producing and detecting high-efficiency stable homozygous mutations and the ability to mutate multiple targeted genes simultaneously is a real benefit of CRISPR for functional genomics in plants [52]. In the present study, we used Cas9 as a single guided RNA to generate targeted rapeseed mutations and to support a stable mutation transmission across the generations (Appendix A). In comparison to Pubi, we used the U6-26p promoter, which showed a higher Cas9 epigenetic modification and editing ability in rapeseed, which contradicted a previous study [44] that suggested the Pubi promoter in dicot plants for Cas9 proteins than U6-26p [53]. The efficient knockout of *bnarfl11* rapeseed homologs via the CRISPR/Cas9 system induced male sterility and similar observations in *Arabidopsis* and demonstrated the use of sgRNA via the CRISPR/Cas9 system to induce targeted mutations in the most critical *Brassica* crop cultivation traits based on the knowledge of the model plant gene function. This study evaluated the editing accuracy of a single copy of BnaRFL11. We examined 15 possible off-target loci from T_0_ mutant plants, none of which showed induced CRISPR/Cas9 mutations, suggesting that well-designed specific sgRNA does not target any other site, and there is a marginal risk of off-target effects (Appendix A).

Several main factors affect the sgRNA potency in plants, including Cas9 and sgRNA expression values, G.C. content %, targeting circumstances, and the key targeted structure of sgRNAs [53]. In the present study, targeted deletions in most transgenic positives, S1 and S2, resulted in high levels of single-target DSBs. We also discovered that the G.C. content of the current study sgRNAs ranged from 45% to 65%, optimal for sgRNAs, and the mutagenesis performance ranged from 0% to 65.10% at T_0_ for the two sgRNAs (Appendix A), suggesting that almost all the promoters were involved in rapeseed genome editing. The effectiveness of the target site, the guiding RNA design, the adequacy of the Cas9 expression, and the inter-analysis of the transformants for more numerous sgRNA-based plant editing are all factors that limit the editing performance. The editing efficacy of sgRNA (S1–S2) has been evaluated using a single copy of *the bnarfl11*. These sgRNAs demonstrated a dramatic efficiency in genome editing for targeting the same gene, such as 64.81% at S1 and 46.29% at S2. We also compared the editing efficiency of sgRNA in our present study with that of the previous study. [54] targeted three genes, *BnCLV1*, *BnCLV2*, and *BnCLV3*, with ten different sgRNA driven by the same promoter. [40] also targeted three genes, *BnIND*, *BnALC*, and *BnTT8*, using 12 different sgRNAs driven by the same promoter. The data in (Appendix A) represent that the editing efficiency of the U6-26p promoter was higher than that of the other promoters. In short, variations in the efficacy of the genome editing of several sgRNAs most likely indicate changes in the nucleotide composition of the sgRNAs.

### 4.2. Precise Identification of Allelic Variation with T.A. Clone following Sequencing

A PAGE-based genotyping method for analyzing CRISPR/Cas9-related mutagenesis at the targets of *bnarfl11* was used in the current study. This helped develop a highly efficient and straightforward mutational detection method (Appendix A). Although the time efficiency of the PAGE-based genotyping method is evident, there is still a limited use, such as small nucleotide indels or substitution replacements in homozygous mutant plants, which are very difficult to detect [46]. T_0_ validated the results of the PAGE screening, we used the high-throughput tracking of the T. A. clone. A total of 88 targeted mutant lines were identified in the T_0_ generation using Sanger sequencing. In both mutant lines, various mutations were detected, including inserting and deleting different nucleotides at the S1 and S2 target sites (Figure 4). Thus, the TA clone-based sequencing method for mutation detection is an efficient and straightforward method with a high mutation frequency. Seventeen mutant lines were selected out of 88 for further generation mutation analyses (Appendix A, Appendix A).

For the successful use of *B. napus*, it is essential to recognize desirable allelic variations and candidate genes for male sterility and proper leaf growth for photosynthesis in China’s tri-annual crop rotation systems. Male sterility is a complex quantitative trait. Male sterility is induced only by mitochondrial ORFs when *Rf,* a nuclear gene, fails to diminish the *ORF* function. In the current study, *BnaRFL11* triple and quadrable homozygous mutants had severe male sterility and an abnormal leaf phenotype (Figure 4, Figure 5a–f, Figure 8a–p and Figure 9a–d). CRISPR/Cas9 has not yet developed impulsive or induced male sterility mutants in *B. napus*.

### 4.3. BnaRFL11 Gene Plays an Essential Role in Fertility Restoration in B. napus

*Rf* genes are well known in *Arabidopsis* and other crops, as they restore male fertility. *RFL* is a class of *PPR-*encoding genes; *RFL* genes have functions similar to PPR in RNA modification, stabilization, and cleavage. Thus, *RFL* is a successful *Rf* candidate that could play a key role in fertility restoration. However, we still have limited knowledge of the *RFL* genes in other species, such as rapeseed. In a previous study, 53 *BnaRFLs* were identified, of which *BnRFL13* (*Rfp*) and *BnaRFL6* (*Rfn*) were found on chromosome A9 inside the gene cluster [29]. The first study on the loss of the *RFL* mutant in *Arabidopsis* showed that the *AtRFL2* mutant had an abnormal phenotype [55].

*RPF8* is an *RFL*-PPR protein [21,56]. A linkage study validated the involvement of *the RFP8* protein in producing the −141 5′ ends of the *nad3-rps12* mRNA in Van−0 [57]. The endonucleolytic cleavage of the mitochondrial gene *orf291*, which prevents transcript accumulation, is mediated by the *Arabidopsis* mitochondrial protein *RFL2*. *PRORP1*, a proteinaceous protein involved in mitochondrial RNA processing, facilitates this cleavage [48,58]. *RPF8* and *RFL2* from *A. thaliana* were similar to a single copy of *BnaRFL11* (Figure 1 and Figure 2).

In the current study, all homozygous variations of the T_0_ and T_1_ lines were grown in the Hubei province to characterize *the BnaRFL11* mutant phenotypes. Compared to wild-type plants, all transgenic lines showed a dramatic change in the floral structure with stable male sterility, as explained previously [59]. Microsporocytes and other locule features are uncommon, and all sites within CMS anthers rarely fail to form, as shown by nap CMS in rapeseed. Because of the disturbance of the anther development symmetry and the polarity of the developmental anomalies, the locules proximal to the carpel have a much greater risk of forming than locules distal to the carpel; as a result, they are among the additional and hyper sexualized features of the morphology of *Brassica* CMS. The cytological analysis (Figure 8) and expression analysis by qRT-PCR (Figure 10) of *orf222/orf139/nad5c* in sterile transgenic lines confirmed that the anther abortion stage is closely similar to the nap-CMS type. The dramatic rise in mitochondrial transcripts, particularly in anther tissues in CMS plants, also includes copies of the CMS-related gene *orf222*. The N-terminal encoding segment of *orf222* is provided by atp8, which encodes a cellular ATP synthase moiety. As recommended by [60], by interfering with the function of its regular counterpart, the expression of a CMS-related gene product that mimics a regular mitochondrial product can cause mitochondrial damage. ATP8 is essential for ATP synthase aggregation. It is conceivable that in rapeseed CMS, a dysfunctional synthase array throughout phases of the successful mitochondrial function in sporogenous tissue causes oxidative damage that is severe enough to disrupt the average anther development [61]. *AP3* and other B function genes regulate the identification and functional symmetry of the stamen and petals. According to our findings, an *AP3* predictor array’s CMS activity timing differs from that of fertile stamens and petals.

As a result, the specific elements of CMS morphology that we examined here, comprising both stamen and petal modifications, may be caused by changes in the *AP3* function. Our findings indicate that early in stamen growth, the cell-specific expression of *orf222* prevents sporogenesis from starting at specific sites, resulting in sterile anthers. Since the *Rfn* gene failed to restore fertility, our findings suggest that this consequence could be regulated by *orf222* activity on the output of the nuclear genes involved in anther processing, such as *AP3*.

Flowering plants have also developed a unique method for controlling organelle gene expression, in which post-transcriptional courses play a significant role in determining gene stoichiometry. The pentatricopeptide repeat function (PPR) proteins in plant RNA research have been among the most significant breakthroughs in the last few decades. Restorer of fertility (*Rf*) genes are nuclear loci suppressing CMS development. *The Rf* genes also encode pentatricopeptide repeat proteins. *RFL* genes have been implicated in the post-transcriptional alteration of mito-RNA in many studies. In *Arabidopsis*, pentatricopeptide repeat (*PPR*) protein RNA processing factor8 *(RPF8*) and *RFL2* are involved in the 5′ processing of various mitochondrial mRNAs. Both factors are similar to *Rf*, found in various plant species’ CMS/restoration systems. These results indicate that *Rf*-like *PPR* proteins play a crucial role in post-transcriptional 5′-processing. Collectively, the *RFL* gene family provides a new way to explore restorer genes in other CMS systems to complement traditional genetic mapping to locate the candidate genes. The mutants of *BnaRFL11* developed in the current study might help researchers further investigate the molecular mechanism of *RFL* genes in fertility restoration and Mendelian inheritance in hybrid breeding (Figure 11).

## 5. Conclusions

The present study utilized the CRISPR/Cas9 system to knock out a single copy of the *BnaRFL11* gene (*BnaA09g45590D*) to develop plants with male sterility in *B. napus.*

(1)In a single copy of *bnarfl11*, heritable mutant plants develop different mutant lines with a constant mutation transmission in the T_0_ and T_1_ generations. Free T-DNA mutant lines were generated across two successive generations.(2)The phenotypic analysis of *bnarfl11* revealed defects in the floral structure, leaf size, branch number, and seed production. The phenotypic analysis showed that the mutant plant lines had male sterility with multiple branches and a reduced leaf size compared to the W.T. plant lines.(3)To determine whether the off-target effect occurred in this study, we used the “CRISPR-P” software to identify a supposed off-target site in the rapeseed genome that is homologous to the two designed sgRNAs. The locations of potential off-target sites in the genome were noted. According to the sequencing (TA-Clone) of the PCR products, seven possible off-target sites from several T_0_ mutated plants showed no mutations.(4)The editing efficiency of the two sgRNAs of the *bnarfl11* gene ranged from 0% to 65.10% in T_0_. We analyzed the editing efficiency of each sgRNA (S1–S2) that targets the *bnarfl11* gene.

The current study’s findings suggest that CRISPR/Cas9 mutations may divulge the functions of genes in polyploid species and provide agronomically desirable traits through a targeted mutation.

## 6. Future Prospective

Future work can be conducted in the following ways.

(1)To further identify the signaling pathway and molecular mechanism of *BnaRFL11* in controlling other related genes involved in male sterility.(2)The development of markers for the polymorphic identification of variations in different mutation sites for CMS breeding in *B. napus*.

## Figures and Tables

**Figure 1 plants-11-03501-f001:**
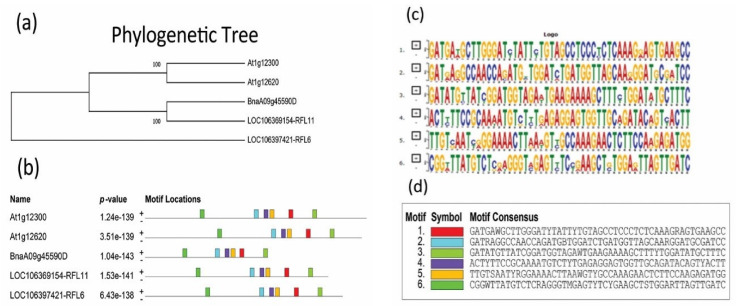
Conserved motif analysis and phylogenetic relationship of *BnaRFL11* with other gene species. (**a**) Phylogenetic tree showing an association between *BnaRFL11* and its sister genes in *Arabidopsis*. Protein sequences were obtained from the GenBank with the following gene ID: *BnaA09g45590D*, *AT1G12300*, and *AT1G12620.* (**b**) Presence of similar motifs between sequences. (**c**,**d**) The conserved motifs among *BnaRFL11,* known fertility restore genes *Rfn* and *Rfp*, and *Arabidopsis thaliana* sister genes.

**Figure 2 plants-11-03501-f002:**
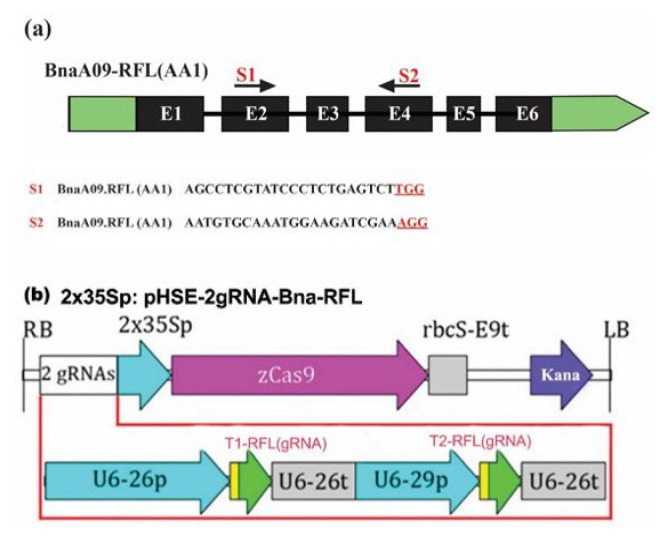
Model of *BnaRFL11* with target sequences and the binary plasmid Cas9 vector. (**a**) The *BnaRFL11* gene structure is shown in black. The vertical line indicates the gene model, and the arrow shows the sgRNA direction. The target sequences of sgRNAs are represented by PAM sites highlighted in red. (**b**) The kanamycin resistance cassette is driven by the CaMV35S promoter of the cauliflower mosaic virus. Two sgRNAs are driven by the U6-26p and U6-29p promoters and the U6-26t terminator.

**Figure 3 plants-11-03501-f003:**
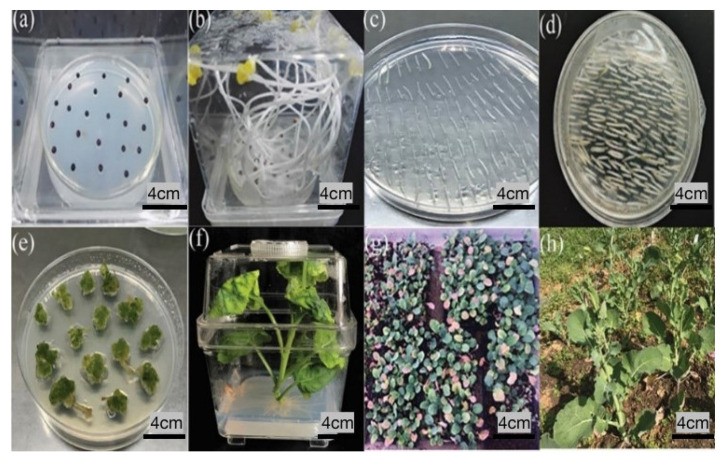
Different stages of the tissue culture process for Agrobacterium-mediated transformation in rapeseed. (**a**) Sowing of wild-type Westar seeds on the seedling medium (M0). (**b**) Growth of plants on seedling medium after 6–7 days. (**c**) Spreading of hypocotyls for 48 h in the dark on (M1) media. (**d**) After 48 h, transfer explants were transferred to (M2) media and placed in a growth room for 15–20 days. (**e**) After 15–20 days, explants were on (M3) media for callus induction until the plants were regenerated. (**f**) Regenerated plants were transferred into (M4) rooting media to obtain better roots. (**g**) After obtaining roots, transgenic plants were transferred into pots under controlled conditions for their survival. (**h**) Transgenic plants were transferred to the field under normal conditions after survival. Bar = 4 cm.

**Figure 4 plants-11-03501-f004:**
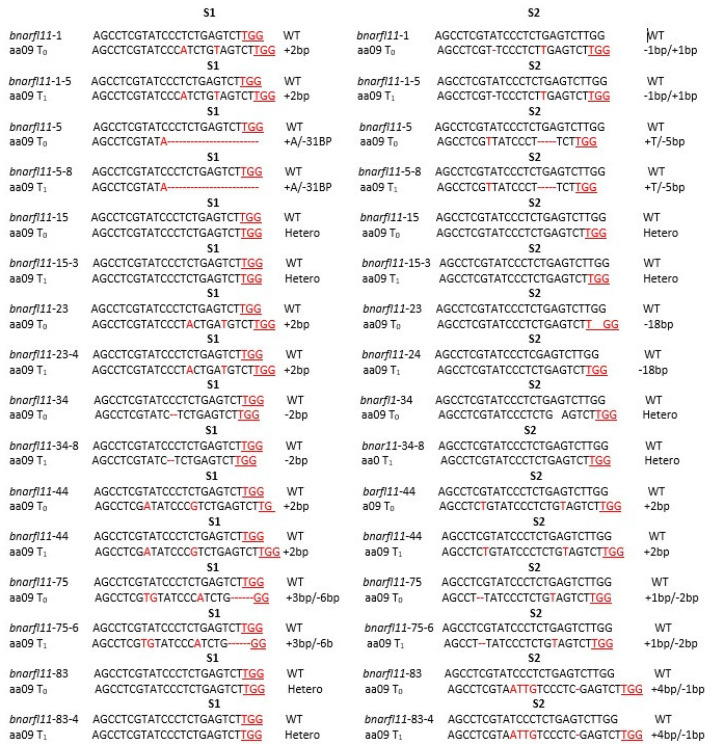
Mutation transmission from T_0_ to T_1_ generations. At both target sites in *bnarfl11* mutant plants. Red font and red hyphens signify CRISPR-based alterations, whereas PAM is underlined and highlighted in red. The altered alleles of *RFL11* on chromosome A were expressed as aa09. “−” and “+” signify losses.

**Figure 5 plants-11-03501-f005:**
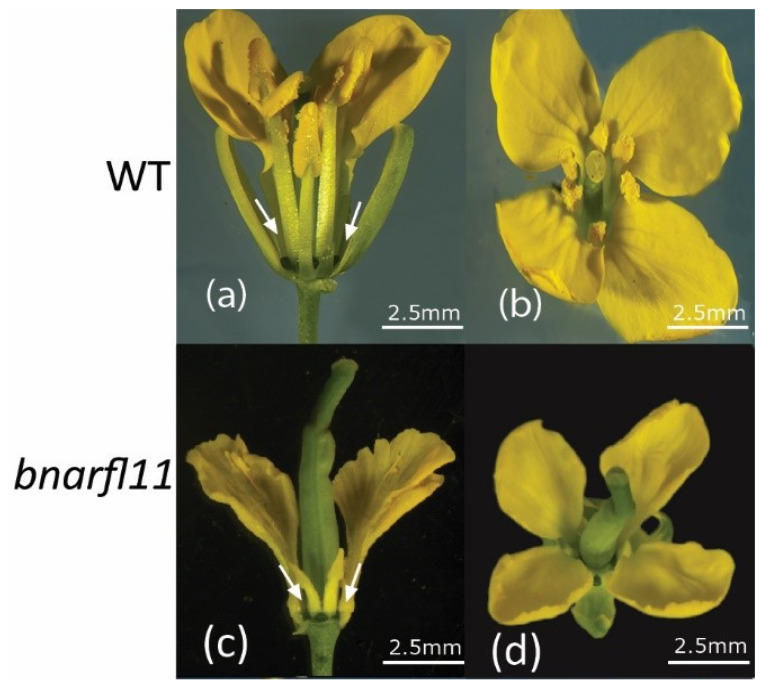
Phenotypic analysis of CRISPR/Cas9 mutated *bnarfl11* plants. (**a**,**b**) Shows wild type with normal flower development. (**c**,**d**) Shows *bnarfl11* flower.

**Figure 6 plants-11-03501-f006:**
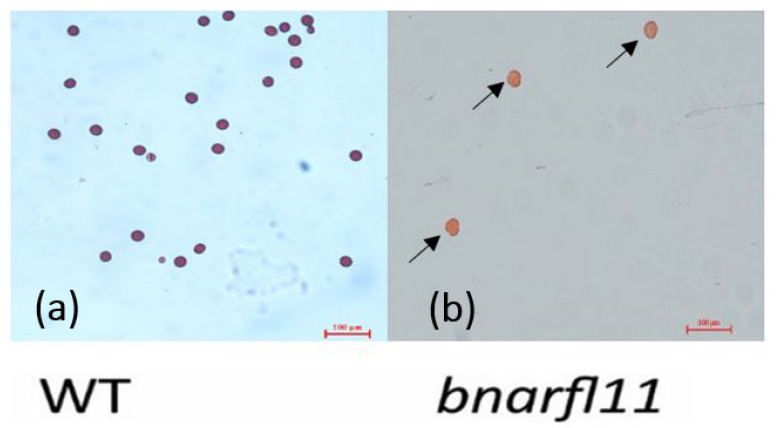
Pollen viability test staining with 1% (*w*/*v*) acetocarmine solution. (**a**) Shows WT pollens. (**b**) Arrows indicating *bnarfl11* degenerated pollens.

**Figure 7 plants-11-03501-f007:**
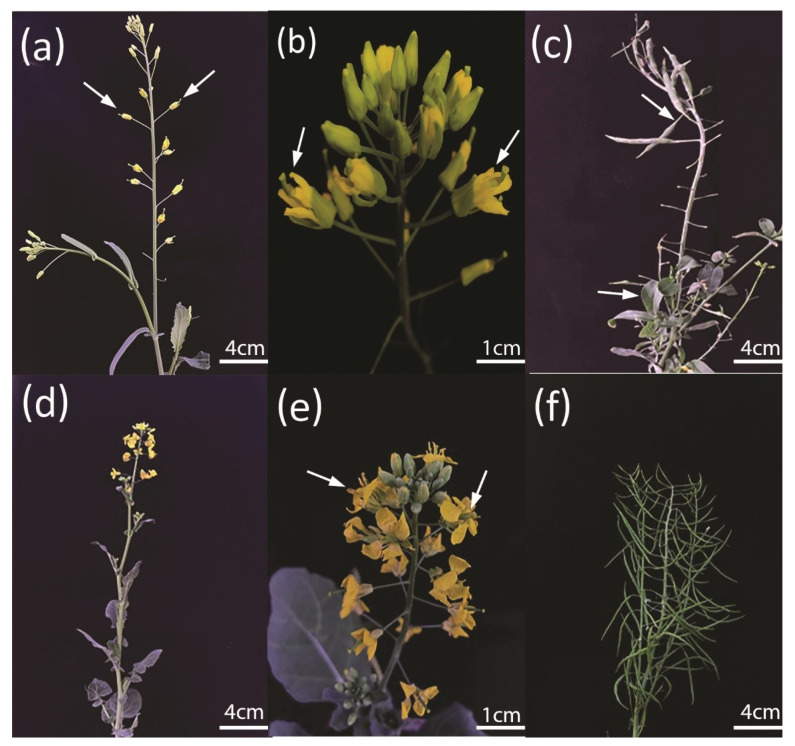
Plant growth stages of *bnarfl11* compared to W.T. (**a**) Male-sterile plant at the close flower stage. (**b**) Male-sterile plant at the open-flower stage. (**c**) Male-sterile plants after crossing with the wild type with significantly fewer seeds. (**d**–**f**) Wild-type growth stages compared with those of *bnarfl11*.

**Figure 8 plants-11-03501-f008:**
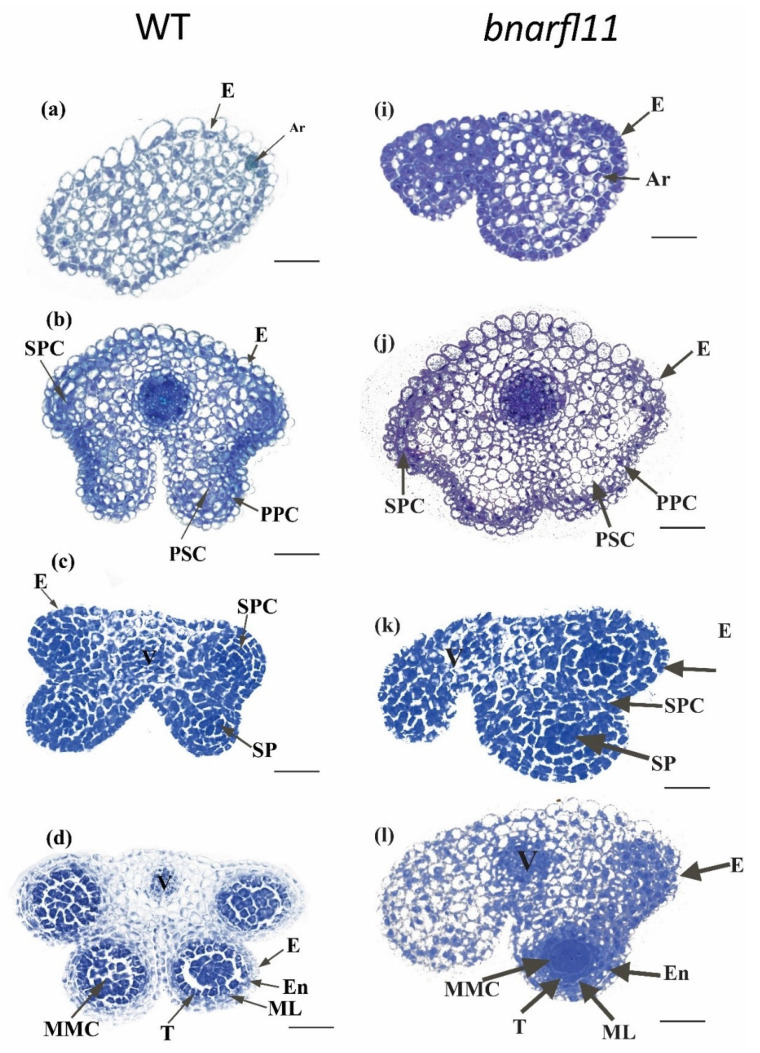
Cytological analysis of *bnarfl11* (**i**–**p**) compared to the wild type (**a**–**h**). There is no significant difference at early stages (**a**,**b**,**i**,**j**), but at later stages (**c**,**d**,**k**,**l**), the loss of one or two locules per anther is evident. In (**l**), MMCs are intermittently formed with an intact tapetum in the locks compared to W.T. **Ar**, archesporial cell; **E**, epidermis; **PSC**, primary sporogenous cell; **MMC**, microspore mother cells; **S.P.**, sporogenous cell; **En**, endothecium; **SPC**, secondary parietal cell; **ML**, middle layer; **T**, tapetum; V, vascular region; **PPC**, primary parietal cell Bar = 10 μm. Cytological analysis of *bnarfl11* compared to the wild type. (**e**–**h**) Stages of the wild type from **5–8**. (**m**–**p**) displays the stages of the sterile flower from **5–8**. (**m**) The Meiocyte stage with a highly dense callose wall around them and vacuolation in the tapetum was more confirmed than W.T. (**e**,**n**) At the tetrad stage, clumps of dense tissue in CMS locules, while in W.T. (**f**) locules and MC are typical. (**o**) At the pre-dehiscent stage, the locule increases in size, microspores form a cell clump, and the tapetum starts degrading; however, in W.T. (**g**), they have four large locules and normal microspores. (**p**) Degenerated pollen grains in the CMS flowers. (**h**) Normal pollen grains. **T**, tapetum; **MC**, meiotic cell; **E**, epidermis; **St**, stomium; **Msp**, microspores; **S.P.**, sporogenous cell; **Tds**, tetrads; **V**, vascular region; **P.G.**, pollen grain; **DGP**, degenerated pollen grains Bar = 10 μm.

**Figure 9 plants-11-03501-f009:**
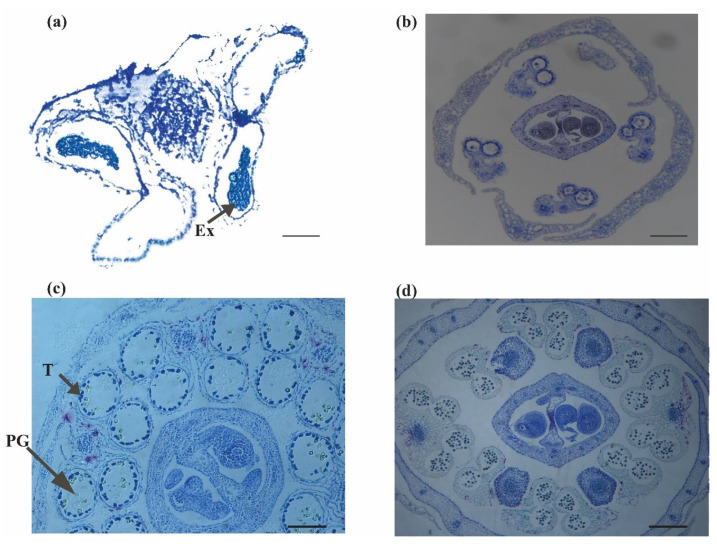
Dehiscence stage of *bnarfl11* compared with W.T. (**a**,**b**) *bnarfl11* dehiscence stage. (**a**) Heap of scarring in exine (Ex) if four locks succeed in development. (**b**) The loss of adaxial or abaxial locules is evident in *the bnarfl11*. (**c**,**d**) Wild-type flowers at dehiscence, with four locules and normal mature pollen grains. Bar = 10 μm.

**Figure 10 plants-11-03501-f010:**
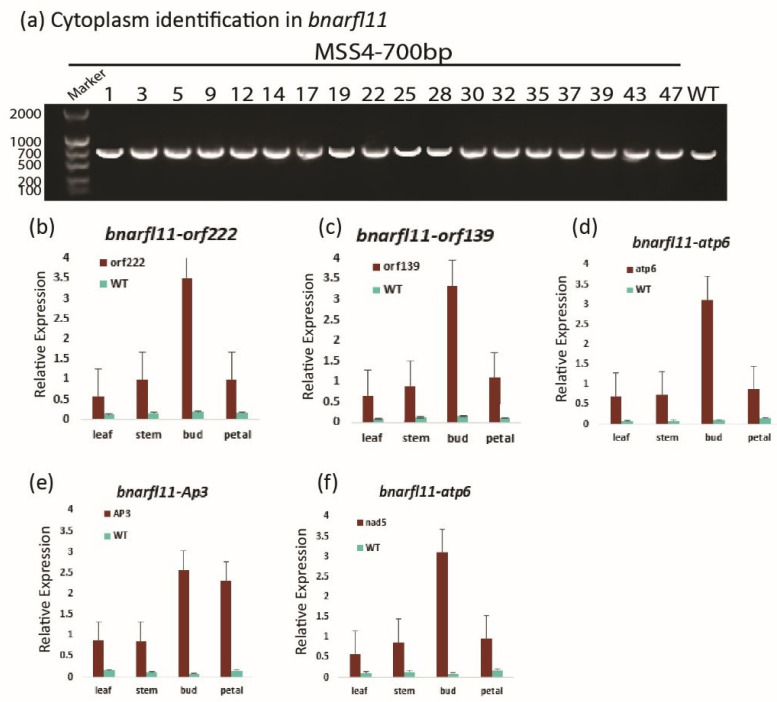
RT-PCR and qRT-PCR of *bnarfl11* plants. (**a**) RT-PCR analysis of nap-CMS cytoplasm in *bnarfl11* plants. For RT-PCR, total DNA was extracted from the Cas9 transgenic line, and the W.T. leaves. (**b**–**f**) qRT-PCR analysis of CMS-causing genes in *bnarfl11* plants. For qRT-PCR analysis, total RNA was extracted from leaves, stems, buds, and petals of the Cas9 transgenic line. Values represent the mean and standard deviation. Error bars indicate the standard deviation among triplicate experiments.

**Figure 11 plants-11-03501-f011:**
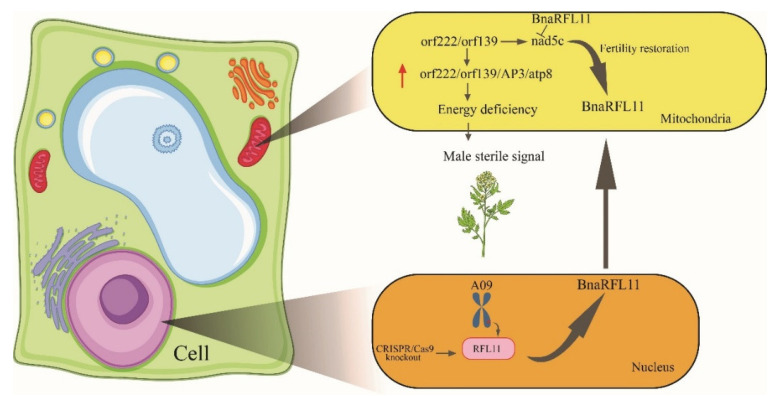
Putative model for male sterility in the current study on rapeseed. Red arrow indicating upregulation.

**Table 1 plants-11-03501-t001:** Statistical results for *the bnarfl11* transgenic generations.

Target Gene	Number of Targets	Generation	Transgene Rate (%)	Positive Rate (%)
*BnaRFL11*	2	T_0_	81% (88/108)	89% (97/108)
	2	T_1_	75% (110/145)	90% (131/145)

Numbers in brackets indicate the number of mutated plants divided by the total number of plants developed.

## Data Availability

All data generated or analyzed during this study are included in this published article (and its Appendix A).

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
