# Peer review of "Induction of Male Sterility by Targeted Mutation of a Restorer-of-Fertility Gene with CRISPR/Cas9-Mediated Genome Editing in Brassica napus L."

_plants, 2022, doi:10.3390/plants11243501_

Round 1
Reviewer 1 Report
The manuscript is dealing with the induction of male sterility by CRISPR/Cas9-mediated targeted mutation of a putative fertility restorer gene in Brassica napus. The paper deploys a broad range of experimental instrumentation and provides interesting results.
However, the paper is poorly written, its language is pale and the manuscript contains unclear sentences, grammatical and wording errors. Such weaknesses cannot be tolerated in a high profile scientific journal. In its present form the manuscript can at most be considered as an early-level draft, which needs thorough editing, re-structuring and styling.
The Abstract (which is likely to be taken as the first piece of information by most of the readers) is difficult to read and interpret and is not focusing at the most relevant results and achievements.
Here there are two consecutive sentences from the Abstract (Lines 26-28):
"Restorer of fertility-like genes (RFL) by developing targeted mutations in polyploid species of rapeseed (B. napus). Characterize the mutants' molecular identity, including mutation sites, mutation types (insertion, deletion, and substitution)."
One can conclude that these sentences are intended to be "bullet points" of a listing of objectives. However, such listings are not common (and not preferred) in an Abstract, but it is expected that every sentences will be meaningful and grammatically correct. And these two sentences aren't ...
Further, the authors create confusion by not clearly distinguishing between Restorer of fertility (Rf) and Restorer of fertility-like (RFL) genes in the Abstract:
"...fertility restorer (Rf) genes." - Line 21 ;
"...restorers of fertility (RFL) genes ..." - Line 22.
The main parts of the paper are again full of sloppy, carelessly formulated or even scientifically incorrect sentences. Just a few examples:
Lines 49-51:
"The plant male sterility is beneficial as a common biological phenomenon in research on another development, cytoplasmic origins, nucleo-cytoplasmic interactions."
(What does "research on another development" mean ???)
Lines 55-56:
"Mitochondrial MS is an essential function of cell genomes that have been evolved and used to breed rapeseed..."
(Why is mitochondrial MS an "essential function ??? What does "cell genomes" mean ???)
Lines 61-62:
"PPR proteins are mainly encoded by Rf genes ..."
This statement is definitely false. PPR proteins represent one of the largest plant protein family (with over 400 members in most of the sequenced genomes) of diverse functions. A relatively small subset of PPR proteins represent Restorer of fertility-like (RFL) genes and so far a few of the RFL genes have been identified as true Restore of fertility (Rf) genes.
Lines 89-90:
"Rapeseed has emerged as a model plant for researching the effects of polyploidization on passage details and long-distance signaling, based on agricultural studies."
An imprecise and fuzzy sentence again. (No further comment.)
Line 70:
"1.1. Research Progress of Predecessors"
This was chosen as a sub-section title within the Introduction. It doesn't sound so well, though.
In my interpretation, the main message of the paper can be summarized as follows:
(i): Based on sequence homology to an Arabidopsis reference sequence, the authors identified 53 RFL candidate genes in B. napus;
(ii): Of the 53 RFL homologs, one sequence, BnaRFL11 was chosen as a putative Restorer of fertility (Rf) gene;
(iii): CRISPR/Cas9-based inactivation of the BnaRFL11 gene caused male sterility in the knock-out lines.
While the last (iii) finding can be considered as a novel and interesting discovery, some of the follow-up conclusions and interpretations of the authors are vulnerable and giving reasons for criticisms:
Cytoplasmic male sterility (CMS) is caused by the dysfunction of mitochondrial genes and show maternal inheritance. Nuclear Restorer of fertility (Rf) genes can suppress the male-sterile phenotype and restore the production of fertile pollen
The P subfamily of PPR proteins (the class that includes Rf proteins) is generally believed to take part in RNA stabilization and translational activation, which are essential functions for building functional gametes.
Thus, a deleterious mutation in any of the genes of this class might lead to reduced or missing fertility. On the other hand, the authors of this manuscript are lacking the evidence that their candidate gene BnaRFL11 is a functional Rf gene.
(Sequence similarity to a known Rf gene isn't a satisfying evidence for that.)
The male-sterile trait of the lines produced in this project is supposed to follow Mendelian inheritance and it will be impossible to maintain such lines in homozygous state - therefore, such lines will not be suitable to use for mass-production of hybrid seeds.
Consequently, the statement of the authors that "The mutants of BnaRFL11 developed in the current study might help hybrid breeding by CMS in rapeseed" (Lines 543-544), sounds somewhat too optimistic.
To sum it up, I recommend that the authors carefully edit the present manuscript, re-consider and re-interpret some of results and re-submit and improved and styled version in the near future.
Author Response
Dear reviewer,
Thank you very much for your comments concerning our manuscript entitled “Induction of Male Sterility by Targeted Mutation of the RFL Gene with CRISPR/Cas9-Mediated Genome Editing in
Brassica napus L.” plants-1939971. These comments are all valuable and very helpful for revising and improving our paper, as well as the essential guiding significance to our research. We have studied the comments carefully and made corrections, which we hope meet with approval. The response to your comments is as follows:
Response to reviewer#1
The manuscript deals with the induction of male sterility by CRISPR/Cas9-mediated targeted mutation of a putative fertility restorer gene in Brassica napus. The paper deploys a broad range of experimental instrumentation and provides interesting results.
The Abstract (which is likely to be taken as the first piece of information by most of the readers) is difficult to read and interpret and is not focusing at the most relevant results and achievements.
- Here there are two consecutive sentences from the Abstract (Lines 26-28): “Restorer of fertility-like genes (RFL) by developing targeted mutations in polyploid species of rapeseed (B. napus). Characterize the mutants’ molecular identity, including mutation sites, mutation types (insertion, deletion, and substitution).” One can conclude that these sentences are intended to be “bullet points” of a listing of objectives. However, such listings are not common (and not preferred) in an Abstract, but it is expected that every sentence will be meaningful and grammatically correct. And these two sentences aren’t...
Response: Thank you very much for your suggestions. Our description is very fuzzy and not written. Considering this problem, we have deleted this sentence and rewritten the whole abstract in clear writing.
- Further, the authors create confusion by not clearly distinguishing between Restorer of fertility (Rf) and Restorer of fertility-like (RFL) genes in the Abstract: “...fertility restorer (Rf) genes.” - Line 21 ; “...restorers of fertility (RFL) genes ...” - Line 22.
Response: Thank you very much for your suggestions. Now we have made a clear difference between Rf-PPR (Fertility restorer gene) and Rf-like (RFL) Line 27-32.
Rf-like (RFL) genes and Rf-PPR (Rf) genes are homologs and share traits such as high non-synonymous/synonymous codon replacement ratio (dN/dS), independent gene duplications, identification of RNA interfaces, and involvement in post-transcriptional editing of mitochondrial RNA in crops. Since 2011, many researchers have quarried Rf-like (RFL) genes in Arabidopsis thaliana due to their high similarity with Rf genes.
- Lines 49-51: “The plant male sterility is beneficial as a common biological phenomenon in research on another development, cytoplasmic origins, nucleo-cytoplasmic interactions.” (What does “research on another development” mean ???)
Response: Thank you very much for your suggestions. This sentence has no sagacity, so we rewrite this Line 58-59.
Due to mitochondrial impairment, cytoplasmic male sterility offers unique mechanisms to reveal plants’ genetic association between nuclear genomes and mitochondria.
- Lines 55-56: “Mitochondrial MS is an essential function of cell genomes that have been evolved and used to breed rapeseed...” (Why is mitochondrial MS an “essential function??? What does “cell genomes” mean ???)
Response: Thank you for pointing out that this was not clear. This sentence has no understanding, so we rewrite this Line 61-62.
In many plants, CMS traits are determined by ORFs encoded by the mitochondrial genome. Typically, rapeseed has two types of CMS mitotypes named nap and pol.
- Lines 61-62: “PPR proteins are mainly encoded by Rf genes ...” This statement is definitely false. PPR proteins represent one of the largest plant protein family (with over 400 members in most of the sequenced genomes) of diverse functions. A relatively small subset of PPR proteins represent Restorer of fertility-like (RFL) genes and so far a few of the RFL genes have been identified as true Restore of fertility (Rf) genes.
Response: Thank you very much for your suggestions. However, we rewrote this sentence and added more information. Our main point in this line was that most of the Rf genes in crops encode the PPR protein family. It may be due to Rf-PPR having a common ancestor. Line 68-69, Line 73-79.
- Lines 89-90: “Rapeseed has emerged as a model plant for researching the effects of polyploidization on passage details and long-distance signaling, based on agricultural studies.” An imprecise and fuzzy sentence again.
Response: Thank you for your suggestion. We want to delete “Rapeseed has emerged as a model plant for researching the effects of polyploidization on passage details and long-distance signaling, based on agricultural studies” and revise it to.” It can be concluded that six BnaRFL as a candidate for fertility restorer genes (RFL3, RFL4, RFL5, RFL8, RFL15, and RFL41), four 90% similarity with restorer genes (RFL2, RFL10, RFL11, and RFL42), and two restore genes (Rfn-RFL6 and Rfp- RFL13) clustered together in the phylogenetic tree, indicating that these genes were the most likely restorer gene members in the CMS rapeseed model. Line 92-93
- Line 70: “1.1. Research Progress of Predecessors” This was chosen as a sub-section title within the Introduction. It doesn’t sound so well, though.
Response: Thank you very much for your suggestions. First, according to your advice, we have merged this with the Introduction rather than a separate heading.
Minor comments; In my interpretation, the main message of the paper can be summarized as follows:
(i): Based on sequence homology to an Arabidopsis reference sequence, the authors identified 53 RFL candidate genes in B. napus;
(ii): Of the 53 RFL homologs, one sequence, BnaRFL11 was chosen as a putative Restorer of fertility (Rf) gene;
(iii): CRISPR/Cas9-based inactivation of the BnaRFL11 gene caused male sterility in the knock-out lines.
While the last (iii) finding can be considered as a novel and interesting discovery, some of the follow-up conclusions and interpretations of the authors are vulnerable and giving reasons for criticisms:
Cytoplasmic male sterility (CMS) is caused by the dysfunction of mitochondrial genes and show maternal inheritance. Nuclear Restorer of fertility (Rf) genes can suppress the male-sterile phenotype and restore the production of fertile pollen. The P subfamily of PPR proteins (the class that includes Rf proteins) is generally believed to take part in RNA stabilization and translational activation, which are essential functions for building functional gametes. Thus, a deleterious mutation in any of the genes of this class might lead to reduced or missing fertility. On the other hand, the authors of this manuscript are lacking the evidence that their candidate gene BnaRFL11 is a functional Rf gene. (Sequence similarity to a known Rf gene isn’t satisfying evidence for that.). The male-sterile trait of the lines produced in this project is supposed to follow Mendelian inheritance and it will be impossible to maintain such lines in homozygous state - therefore, such lines will not be suitable to use for mass-production of hybrid seeds. Consequently, the statement of the authors that “The mutants of BnaRFL11 developed in the current study might help hybrid breeding by CMS in rapeseed” (Lines 543-544), sounds somewhat too optimistic. To sum it up, I recommend that the authors carefully edit the present manuscript, re-consider and re-interpret some of results and re-submit and improved and styled version in the near future.
Response; Thank you for your kind comment and detailed suggestions. We have revised the manuscript carefully and added some helpful information about BnaRFL11 sequence similarity with the known Rf gene. Please consider our study as a prime discovery, and more work can be done based on these findings, like whether or not the molecular mechanism of RFL in fertility restoration and weather is useful in hybrid breeding. You are correct in line 543-544. The sentence is optimistic without any solid evidence, so we rewrite it as “The mutants of BnaRFL11 developed in the current study might help researchers to dig more about the molecular mechanism of RFL genes in fertility restoration and Mendelian inheritance in hybrid breeding”.
Reviewer 2 Report
Potentially inetersting story, but very difficult to read because of so much long sentences with the lack of logic,
Line 36 – 38: the paraffin section can not suggest itself. Please, edit this sentence.
Line 46 – 51: please, re-write more logical way.
Line 71: please, edit.
Line 164-167: very long sentense with so many different messages. Please, separate.
Line 183: „Although Brassica napus has 53 rapeseed RFL (PPR-protein-coding) genes, according to the genome information released BnaA09g45590D the selected gene of B. napus contains one copy... – please, edit
Line 201: rereported?
Lines 225- 231: It looks like number of sentences without strong logic. For example, „We conducted polyacrylamide gel electrophoresis [37]“ does not fit with the sense.
Figure 3.3 ? There are no any scale bars! Please, re-write legends, look carefully on point c, for example. You did not show a transformation process, but hypocotyl after transformation…
Line 249: „Seventy positive plant PCR fragments were sequenced from a company.“ ??? What do you mean here? Did you mean that sequences was done by a company?
Line 321: „For bnarfl11 gene-phenotype analysis, all the homozygous mutants T1 lines were 321 grown in the genetically modified experimental plot of Huazhong Agricultural Univer- 322 sity, Wuhan, China. „ ??? How did you modify experimental plot?
Figure 6: Pollens without labeling are not visible, the numbers can not be counted.
Many grammar corrections are required.
Conclusions are required better formulations.
Author Response
Dear reviewer,
Thank you very much for your comments concerning our manuscript entitled “Induction of Male Sterility by Targeted Mutation of the RFL Gene with CRISPR/Cas9-Mediated Genome Editing in
Brassica napus L.” plants-1939971. These comments are all valuable and very helpful for revising and improving our paper, as well as the essential guiding significance to our research. We have studied the comments carefully and made corrections, which we hope meet with approval. The response to your comments is as follows:
Response to reviewer#2
Potentially inetersting story, but very difficult to read because of so much long sentences with the lack of logic,
- Line 36 – 38: the paraffin section can not suggest itself. Please, edit this sentence.
Response: Thank you very much for your suggestions. Considering this problem, we have deleted this sentence and rewritten lines 41-43.
- Line 46 – 51: please, rewrite more logical way.
Response: Thank you very much for your suggestions. We have deleted this sentence and rewritten lines 51-56.
- Line 71: please, edit.
Response: Thank you very much for your suggestions. We have edited this statement and merged it with the introduction.
- Line 164-167: very long sentense with so many different messages. Please, separate.
Response: Thank you for pointing out that this was not clear. This sentence has no understanding, so we rewrite this Line 177-179.
- Line 183: „Although Brassica napus has 53 rapeseed RFL (PPR-protein-coding) genes, according to the genome information released BnaA09g45590D the selected gene of B. napus contains one copy... – please, edit
Response: Thank you for pointing out that this was not clear. We have edited this statement to Line 193.
- Line 201: rereported?
Response: Thank you for pointing out that this word is reported; accept our apologies. Line 211.
- Lines 225- 231: It looks like number of sentences without strong logic. For example, „We conducted polyacrylamide gel electrophoresis [37]“ does not fit with the sense.
Response: Thank you very much for your suggestions. However, we rewrote this sentence and added more information. Line 254-264.
- Figure 3.3 ? There are no any scale bars! Please, rewrite legends, look carefully on point c, for example. You did not show a transformation process, but hypocotyl after transformation…
Response: Thank you for pointing out that Figure 3.3 is Figure 3; accept our apologies, and for your second question, we have added scales bars, and in Figure 3, we showed all the basic steps for the rapeseed genetic transformation; we edited the statement as follow “Spreading of hypocotyls for 48h under the dark condition on (M1) media”. The word is spreading of hypocotyl on the M1 medium plate.
- Line 249: „Seventy positive plant PCR fragments were sequenced from a company.“ ??? What do you mean here? Did you mean that sequences was done by a company?
Response: Thank you very much for your suggestions. Yes, we have done sanger sequencing from the company after TA cloning the bnarfl11. We also added the required information to the manuscript. Line 258-266.
- Line 321: „For bnarfl11 gene-phenotype analysis, all the homozygous mutants T1 lines were 321 grown in the genetically modified experimental plot of Huazhong Agricultural Univer- 322 sity, Wuhan, China. „ ??? How did you modify experimental plot?
Response: Thank you for pointing out this statement; accept our apologies. This field only specified for genetically modified plants. That is why we wrote this. However, we have changed this statement to lines 333-334.
- Figure 6: Pollens without labeling are not visible, the numbers can not be counted.
Response: Thank you very much for your suggestions. We have edited this picture labeling.
Minor comments; Many grammar corrections are required. Conclusions are required better formulations.
Response; Thank you for your kind comment and detailed suggestions. We have revised the manuscript carefully and added some helpful information in conclusion supporting our main idea that Based on sequence homology to an Arabidopsis reference sequence, and the authors identified 53 RFL candidate genes in B. napus. Of the 53 RFL homologs, one sequence, BnaRFL11 was chosen as a putative Restorer of fertility (Rf) gene. CRISPR/Cas9-based inactivation of the BnaRFL11 gene caused male sterility in the knock-out lines.
Reviewer 3 Report
The manuscript describes the results of an original study on obtaining mutations in a single gene of Brassica napus via CRISPR/Cas9. The manuscript contains all the necessary sections. The research was performed at the high methodological level. The data are illustrated by excellent figures. This study and the conclusions have a clear methodological focus.
However, to my opinion, the statement of the problem is wrong and leads to erroneous affirmations.
1. The material of the research included an inbred line derived from the cultivar Westar(wild type) possessing the fertile type cytoplasm. The idea of the manuscript is connected with cytoplasmic male sterility which is a result of the interaction between CMS inducing mitochondrial genes (orf) and nuclear fertility restoration genes (Rf). Usually, genotypes with fertile cytoplasm are fertile independently on the allelic state of their Rf gene (genes). In fact, the authors obtained the sterility mutations of a nuclear gene which is no not connected in any way with cytoplasmic sterility induced by mitochondrial genes. This is a so-called nuclear sterility that requires changing the statement of the problem and modifications of discussion of the results.
2. The content of the figure 1 is not clear. As follows from bioinformatic search, At1g12300 is a representative of Arabidopsis thaliana Tetratricopeptide repeat (TPR)-like superfamily protein, whereas A1tg12620 is a representative of Arabidopsis thaliana Pentatricopeptide repeat (PPR) superfamily. Is it legal to compare them along as representatives of PPR-protein coding genes (line 183)?
3. In some places the text is written by poor scientific language (for example there many stylistic errors in Abstract and Introduction) and requires careful editing.
I consider that the manuscript reviewed is not yet ready for publication but it can be substantially improved after major revision.
Author Response
Dear reviewer,
Thank you very much for your comments concerning our manuscript entitled “Induction of Male Sterility by Targeted Mutation of the RFL Gene with CRISPR/Cas9-Mediated Genome Editing in
Brassica napus L.” plants-1939971. These comments are all valuable and very helpful for revising and improving our paper, as well as the essential guiding significance to our research. We have studied the comments carefully and made corrections, which we hope meet with approval. The response to your comments is as follows:
Response to reviewer#3
The manuscript describes the results of an original study on obtaining mutations in a single gene of Brassica napus via CRISPR/Cas9. The manuscript contains all the necessary sections. The research was performed at the high methodological level. The data are illustrated by excellent figures. This study and the conclusions have a clear methodological focus. However, to my opinion, the statement of the problem is wrong and leads to erroneous affirmations.
- The material of the research included an inbred line derived from the cultivar Westar (wild type) possessing the fertile type cytoplasm. The idea of the manuscript is connected with cytoplasmic male sterility which is a result of the interaction between CMS inducing mitochondrial genes (orf) and nuclear fertility restoration genes (Rf). Usually, genotypes with fertile cytoplasm are fertile independently on the allelic state of their Rf gene (genes). In fact, the authors obtained the sterility mutations of a nuclear gene which is no not connected in any way with cytoplasmic sterility induced by mitochondrial genes. This is a so-called nuclear sterility that requires changing the statement of the problem and modifications of discussion of the results.
Response: Thank you very much for your suggestions. We have changed the discussion according to your suggestion line 523-537.
- The content of the figure 1 is not clear. As follows from bioinformatic search, At1g12300 is a representative of Arabidopsis thaliana Tetratricopeptide repeat (TPR)-like superfamily protein, whereas A1tg12620 is a representative of Arabidopsis thaliana Pentatricopeptide repeat (PPR) superfamily. Is it legal to compare them along as representatives of PPR-protein coding genes (line 183)?
Response: Thank you very much for your suggestions. First of all, our apologies, but according to TAIR https://www.arabidopsis.org/servlets/TairObject?id=30634&type=locus, information released both genes are PPR-protein family. It further proven by the following published article [1,2]
- In some places the text is written by poor scientific language (for example there many stylistic errors in Abstract and Introduction) and requires careful editing.
Response: Thank you very much for kind response. We have revised the manuscript carefully and added some supportive information.
- Fujii, S.; Suzuki, T.; Giegé, P.; Higashiyama, T.; Koizuka, N.; Shikanai, T. The Restorer‐of‐fertility‐like 2 pentatricopeptide repeat protein and RN ase P are required for the processing of mitochondrial orf291 RNA in Arabidopsis. The Plant Journal 2016, 86, 504-513.
- Schleicher, S.; Binder, S. In Arabidopsis thaliana mitochondria 5' end polymorphisms of nad4L-atp4 and nad3-rps12 transcripts are linked to RNA PROCESSING FACTORs 1 and 8. Plant Mol Biol 2021, 106, 335-348, doi:10.1007/s11103-021-01153-9.
Round 2
Reviewer 1 Report
After accomplishing some changes and corrections, the manuscript changed to the better. However, there are still several problems and weaknesses that keep the manuscript at an unsatisfactory level.
Some minor problems:
Title:
"Induction of Male Sterility by Targeted Mutation of the RFL Gene with CRISPR/Cas9-Mediated Genome Editing in Brassica napus L."
In every plant genomes there are several different Restorer-of-Fertility-Like (RFL) pentatricopeptide repeat containing genes - and as the authors point out in the manuscript, in the B. napus genome 53 RFL genes have been identified. Therefor, referring to a single gene ("the" RFL Gene) without mentioning a unique gene ID in the title is incorrect. A suggestion for an alternative title:
"Induction of Male Sterility by Targeted Mutation of a Restorer-of-Fertility Gene with CRISPR/Cas9-Mediated Genome Editing in Brassica napus L."
Abstract:
Despite of changes, the present Abstract is too long and not focusing to the most important results and achievements. The Abstract should completely be re-written.
Language and style:
The language of the manuscript is still poor, there are still several fuzzy and obscure sentences and the authors often use inadequate scientific terms and/or improper English words. Just a few examples:
Lines 200-201
"The complete sequence alignment showed that polymorphisms differentiate between these genes’ sources ..."
Lines 300-301:
"quadrable mutations"
It is not clear, what the authors mean under this term. Please, clarify this!
Lines 382-383:
"The tapetum is frequently dense and disconnected from the MMC of growing CMS locules."
Beyond the fact that the word "locules" does not exist, the meaning of the of this sentence is completely unclear.
Lines 581-582
"The current study shows that CRISPR/Cas9 mutations can classify gene functions in polyploid species and result in agronomically significant mutations."
A completely windy/meaningless statement. Such sentences should be avoided in serious papers.
I repeatedly would like to point out that the sentences above are just examples and a thorough styling and language editing throughout the whole text is absolutely necessary. The authors should perhaps avail themselves on a professional manuscript editing service.
However, after reading the revised manuscript more carefully, my attention was directed to some facts in the content, which suggest to re-shape the manuscript in a certain way:
Line 196:
"According to the available genomic information, B. napus only has one copy of BnaRFL11."
Lines 492-493:
"In the current study single copy of the BnaRFL11 gene with 100% similarity with BnaRFL6 (Rfn) present on chromosome A09 ..."
Lines 418-419:
"First, we checked the cytoplasm type by using specific primers designed by [51] (Table S1), and we found that our cytoplasm type was nap-cytoplasm (Figure 10a)."
If the above facts are true (and I do not discredit the authors' statements), they imply the followings:
(1) The Brassica napus variety Westar (the variety used for generating the knockout mutants) is a fertility restored CMS line, with nap-type CMS cytoplasm and the line contained a functional nuclear Rf gene gene.
(2) The BnaRFL11 gene which was identified by the authors, is the genuine fertility restorer gene, with shows 100% (?) identity to the previously published BnaRFL6 (Rfn) gene.
(3) Knocking out the BnaRFL11/BnaRFL6 gene resulted in CMS. Which was obviously expected, as BnaRFL11 is not an Restorer-of-fertility-LIKE gene, but a genuine restorer gene.
Again, if these hypothesises can be proven, the whole manuscript (from title to conclusions) should be changed accordingly. However, the authors should provide sronger evidences to support these hypothesises. For example, exact, sequence-level comparison of the BnaRFL11 and BnaRFL6 genes, sequence-level information about the CMS mitochondria of the used lines of Westar origin, or eventually characterization of offsprings from targeted crosses, etc.
In either case, it looks that to round up this project, the authors should make more effort than some text editing and styling that can be accomplished in three days.
I wish them good luck for these.
I am again proposing to reject this paper, for giving the authors the chance of producing manuscript of better quality. I hope that they will understand my intention!
Author Response
Dear reviewer,
Thank you very much for your comments concerning our manuscript entitled "Induction of Male Sterility by Targeted Mutation of the RFL Gene with CRISPR/Cas9-Mediated Genome Editing in
Brassica napus L." plants-1939971. These comments are all valuable and very helpful for revising and improving our paper, as well as the essential guiding significance to our research. We have studied the comments carefully and made corrections, which we hope meet with approval. The response to your comments is as follows:
Response to reviewer#1
Some minor problems are;
- Title: "Induction of Male Sterility by Targeted Mutation of the RFL Gene with CRISPR/Cas9-Mediated Genome Editing in Brassica napus L." In every plant genome there are several different Restorer-of-Fertility-Like (RFL) pentatricopeptide repeat containing genes and as the authors point out in the manuscript, in the B. napus genome 53 RFL genes have been identified. Therefore, referring to a single gene ("the" RFL Gene) without mentioning a unique gene ID in the title is incorrect. A suggestion for an alternative title:
"Induction of Male Sterility by Targeted Mutation of a Restorer-of-Fertility Gene with CRISPR/Cas9-Mediated Genome Editing in Brassica napus L."
Response: Thank you very much for your suggestions. Considering this problem, we have changed the title of the paper. Your suggestion is suitable. Much obliged.
- Abstract: Despite of changes, the present Abstract is too long and not focusing to the most important results and achievements. The Abstract should completely be rewritten.
Response: Thank you very much for your suggestions. We have tried to shorten the abstract, focusing on the main results.
- Lines 200-201 "The complete sequence alignment showed that polymorphisms differentiate between these genes' sources ..."
Response: Thank you very much for your suggestions. Considering this problem, we have deleted this line because this line delivers the same message as the previous Line 194-195.
- Lines 300-301: "quadrable mutations" It is not clear, what the authors mean under this term. Please, clarify this!
Response: Thank you very much for pointing it out. Quadruple mutation means we have observed mutation in one plant with more than three insertions, deletions, or a mixture of both.
- Lines 382-383: "The tapetum is frequently dense and disconnected from the MMC of growing CMS locules." Beyond the fact that the word "locules" does not exist, the meaning of the of this sentence is completely unclear.
Response: Thank you very much for your suggestions. Considering this problem, we rewrite the line but the word locules exist. A locule is each of a number of small separate cavities. [1].
- Lines 581-582 "The current study shows that CRISPR/Cas9 mutations can classify gene functions in polyploid species and result in agronomically significant mutations." A completely windy/meaningless statement. Such sentences should be avoided in serious papers.
Response: Thank you very much for your suggestions. We have deleted this sentence and rewritten lines 579-581. The current study's findings suggest that CRISPR/Cas9 mutations may divulge the functions of genes in polyploid species and provide agronomically desirable traits by targeted mutation.
- Line 196: "According to the available genomic information, B. napus only has one copy of BnaRFL11."
Response: Thank you very much for your suggestions. We have rewritten Line 193-194.
- Lines 492-493: "In the current study single copy of the BnaRFL11 gene with 100% similarity with BnaRFL6 (Rfn) present on chromosome A09 ..."
Response: Thank you very much for your suggestions. Considering this problem, we have shortened sentences 490-493.
- Lines 418-419: "First, we checked the cytoplasm type by using specific primers designed by [51] (Table S1), and we found that our cytoplasm type was nap-cytoplasm (Figure 10a)." If the above facts are true (and I do not discredit the authors' statements), they imply the followings:
- The Brassica napus variety Westar (the variety used for generating the knockout mutants) is a fertility restored CMS line, with nap-type CMS cytoplasm and the line contained a functional nuclear Rf gene.
- The BnaRFL11 gene which was identified by the authors, is the genuine fertility restorer gene, with shows 100% (?) identity to the previously published BnaRFL6 (Rfn) gene.
- Knocking out the BnaRFL11/BnaRFL6 gene resulted in CMS. Which was obviously expected, as BnaRFL11 is not a Restorer-of-fertility-LIKE gene, but a genuine restorer gene.
- However, the authors should provide sronger evidences to support these hypothesises. For example, exact, sequence-level comparison of the BnaRFL11 and BnaRFL6 genes, sequence-level information about the CMS mitochondria of the used lines of Westar origin, or eventually characterization of offspring from targeted crosses, etc.
In either case, it looks that to round up this project, the authors should make more effort than some text editing and styling that can be accomplished in three days.
Response: Thank you for pointing out that we have provided the phylogenetic analysis and Motif analysis between BnaRFL11/BnaRFL6, showing 100% similarity. We also provided clustal consensus sequences in Supplementary Figure S6.
References
- Geddy, R.; Mahé, L.; Brown, G.G. Cell‐specific regulation of a Brassica napus CMS‐associated gene by a nuclear restorer with related effects on a floral homeotic gene promoter. The Plant Journal 2005, 41, 333-345.
Reviewer 2 Report
Thank you, text is better, but so many points need to be improved.
Line 124: it is better to write obtained, not collected.
Line 244: Numerous plants showed various bands that differ from those on non-denaturing PAGE gels in wild-type plants (WT), Plants itself can not
show bands, bands can be shown by proteins.
Figure 3: please, make bar in black color.
Line 255: Transgenic plants were transferred into the field under normal conditions after their survival.???
Lines 340: morphologic differentiation? Maybe differences?
There are two figures 8. Which is correct one?
line 460: "To compare the editing efficiency of the sgRNA of our present study with the previous study of [55] used the same CRISPR/Cas9 constructs with different sgRNA and the same promoters in B. napus. [55] targeted three genes, BnCLV1 (two copies), BnCLV2 (two copies), and BnCLV3 (two copies), with ten different sgRNA driven by the same promoter- " - long sentence, please, make it simple and more correct.
Author Response
Dear reviewer,
Thank you very much for your comments concerning our manuscript entitled “Induction of Male Sterility by Targeted Mutation of the RFL Gene with CRISPR/Cas9-Mediated Genome Editing in
Brassica napus L.” plants-1939971. These comments are all valuable and very helpful for revising and improving our paper, as well as the essential guiding significance to our research. We have studied the comments carefully and made corrections, which we hope meet with approval. The response to your comments is as follows:
Response to reviewer#2
Thank you, text is better, but so many points need to be improved.
- Line 124: it is better to write obtained, not collected.
Response: Thank you very much for your suggestions. Considering this problem, we deleted this word and obtained line 123.
- Line 244: Numerous plants showed various bands that differ from those on non-denaturing PAGE gels in wild-type plants (WT), Plants itself cannot show bands, bands can be shown by proteins.
Response: Thank you very much for your suggestions. We have deleted this sentence and rewritten lines 243-249.
- Figure 3: please, make bar in black color.
Response: Thank you very much for your suggestions. We have made a bar in Figure 3.
- Line 255: Transgenic plants were transferred into the field under normal conditions after their survival.???
Response: Thank you very much for your suggestions. This statement means we can not directly transfer transgenic plants into the field. We should first grow them in small pots, and when they survive in the pots under controlled conditions, then we transfer them into the field so they will grow better.
- Lines 340: morphologic differentiation? Maybe differences?
Response: Thank you for pointing out that this was not suitable. We have changed to differences Line 343.
- There are two figures 8. Which is correct one?
Response: Thank you very much for your suggestions. Both figures are correct. The first picture shows stages 1-4, and the following shows stage 5-8 comparing WT and bnarfl11. To resolve this problem, we have named 8A and 8B.
- line 460: "To compare the editing efficiency of the sgRNA of our present study with the previous study of [55] used the same CRISPR/Cas9 constructs with different sgRNA and the same promoters in B. napus. [55] targeted three genes, BnCLV1 (two copies), BnCLV2 (two copies), and BnCLV3 (two copies), with ten different sgRNA driven by the same promoter- " - long sentence, please, make it simple and more correct.
Response: Thank you very much for your suggestions. Considering this problem, we have deleted this sentence and rewritten lines 467-470.
Reviewer 3 Report
The revised version of the manuscript was substantially improved. The Authors have took into account all comments and have presented a comprehensive. The most problematical Introduction section was significantly changed. Now its content is clear and understandable. Only minor corrections are required.
Lines 37, 38 – Please correct bad expression “….in the CMS mechanism existing on the A9 and C8 chromosomes”.
Line 52 – I advice to add additional key words. Your list is too short and repeat the words present in the tittle that is undesirable.
Lines 64, 65 – Please edit the sentence. In contains false statement on the CMS induction through self-incompatibility.
I recommend to indicate in the Material and Methods section that the variety Westar (which was used in the experiments) possess the nap-type cytoplasm and, correspondingly, the nap-type specific mitochondrial features. . This is especially important for understanding content of the Figure 10 om the cytoplasm identification.
A few technical and stylistic corrections are also necessary.
I consider that the article can be published after abovementioned minor technical and text edits.
Author Response
Dear reviewer,
Thank you very much for your comments concerning our manuscript entitled “Induction of Male Sterility by Targeted Mutation of the RFL Gene with CRISPR/Cas9-Mediated Genome Editing in
Brassica napus L.” plants-1939971. These comments are all valuable and very helpful for revising and improving our paper, as well as the essential guiding significance to our research. We have studied the comments carefully and made corrections, which we hope meet with approval. The response to your comments is as follows:
Response to reviewer#3
Comments and Suggestions for Authors
The revised version of the manuscript was substantially improved. The Authors have taken into account all comments and have presented a comprehensive. The most problematical Introduction section was significantly changed. Now its content is clear and understandable. Only minor corrections are required.
- Lines 37, 38 – Please correct bad expression “….in the CMS mechanism existing on the A9 andC8 chromosomes”.
Response: Thank you very much for your suggestions. We have deleted this sentence and rewritten it in clear writing, Line 36-37.
- Line 52 – I advice to add additional key words. Your list is too short and repeat the words present in the tittle that is undesirable.
Response: Thank you very much for your suggestions. Now we have made a new list of keywords in Line 51; we hope it will be acceptable now the list is as follow;
Keywords: Rapeseed CMS; Rf-like (RFL); CRISPR/Cas9; sgRNA; Cytological study; Genome editing
- Lines 64, 65 – Please edit the sentence. In contains false statement on the CMS induction through self-incompatibility.
Response: Thank you for pointing out that. This sentence has no understanding as self-incompatibility is a natural mechanism, so we rewrite this Line 63-64.
- I recommend to indicate in the Material and Methods section that the variety Westar (which was used in the experiments) possess the nap-type cytoplasm and, correspondingly, the nap-type specific mitochondrial features. This is especially important for understanding content of the Figure 10 on the cytoplasm identification.
Response: Thank you very much for your suggestions. Per your advice, we have written this in the material and methods part Line 122.
- A few technical and stylistic corrections are also necessary.
Response: Thank you for your kind comment and detailed suggestions. We have revised the manuscript carefully and also added some helpful information.
I consider that the article can be published after above mentioned minor technical and text edits.
Round 3
Reviewer 1 Report
After the third circle of revisions, the manuscript still suffers from serious flaws. In spite of some styling and corrections, there are several obscure sentences, spelling- and typo errors.
In case of a re-submission, the authors should take more care about the correct use of scientific terms and conventions and they are advised to avail on a professional manuscript editing service and/or on proof-reading by experienced collegaues.
Further, the authors should make attempts to provide more robust evidences concerning the CMS type (nap-type cytoplasm) of their input material (as suggested in the previous review).
In the abscence of these, the manuscript can not be recommended for publication in MDPI Plants.
Reviewer 2 Report
Nice work, all points were corrected.
Please, line 901: W:T; - may be wt?
Figure 3: plesae, redesign with proper dimension. Some panels were deformed.
Scale bar: it is not nessary to write so large letters like 1cm, etc. You can mention scale bar in legends.